# Deficiency of Adipose Aryl Hydrocarbon Receptor Protects against Diet-Induced Metabolic Dysfunction through Sexually Dimorphic Mechanisms

**DOI:** 10.3390/cells12131748

**Published:** 2023-06-29

**Authors:** Nazmul Haque, Emmanuel S. Ojo, Stacey L. Krager, Shelley A. Tischkau

**Affiliations:** 1Department of Pharmacology, Southern Illinois University School of Medicine, Springfield, IL 62702, USA; nhaque34@siumed.edu (N.H.); eojo74@siumed.edu (E.S.O.); skrager@siumed.edu (S.L.K.); 2Department of Medical Microbiology, Immunology and Cell Biology, Southern Illinois University School of Medicine, Springfield, IL 62702, USA

**Keywords:** adipose–hypothalamic cross talk, adipose–immune crosstalk, diet-induced obesity, glucose sensitivity, diabetes, aryl hydrocarbon receptor, xenobiotics, sex differences

## Abstract

The molecular mechanisms underlying diet-induced obesity are complex and remain unclear. The activation of the aryl hydrocarbon receptor (AhR), a xenobiotic sensor, by obesogens may contribute to diet-induced obesity through influences on lipid metabolism and insulin resistance acting at various sites, including adipose tissue. Thus, our hypothesis was that conditional AhR depletion, specifically from mature adipose tissue (CadKO), would improve high-fat diet (HFD)-induced metabolic dysfunction. CadKO protects mice from HFD-induced weight gain. CadKO females eat fewer calories, leading to increased energy expenditure (EE) and improved glucose tolerance on HFD. Our exploration of adipose tissue biology suggests that the depletion of AhR from adipocytes provides female mice with an increased capacity for adipogenesis and lipolysis, allowing for the maintenance of a healthy adipocyte phenotype. The HFD-induced leptin rise was reduced in CadKO females, but the hypothalamic leptin receptor (LepR) was increased in the energy regulatory regions of the hypothalamus, suggesting an increased sensitivity to leptin. The estrogen receptor α (ERα) was higher in CadKO female adipose tissue and the hypothalamus. CadKO males displayed a delayed progression of obesity and insulin resistance. In males, CadKO ameliorated proinflammatory adipocytokine secretion (such as TNFα, IL1β, IL6) and displayed reduced inflammatory macrophage infiltration into adipose depots. Overall, CadKO improves weight control and systemic glucose homeostasis under HFD challenge but to a more profound extent in females. CadKO facilitates a lean phenotype in females and mediates healthy adipose–hypothalamic crosstalk. In males, adipose-specific AhR depletion delays the development of obesity and insulin resistance through the maintenance of healthy crosstalk between adipocytes and immune cells.

## 1. Introduction

The global obesity epidemic contributes to millions of deaths each year. The mass consumption of calorie-dense diets and increasingly sedentary lifestyles have led to energy imbalances that promote fat deposition and are considered drivers of obesity and associated diseases, including type 2 diabetes. A better understanding of systemic energy regulation is key to combating these diseases, and adipose tissue is a central player. Adipose tissue function is multifaceted, ranging from energy storage to endocrine and immune functions. Adipose function is drastically altered by the chronic ingestion of excess energy, leading to lipid overloading and adipocyte dysfunction, resulting in metabolic disease. White adipose tissue (WAT) is composed of adipocytes, preadipocytes, and immune cells and is surrounded by extracellular matrix and vascular networks [1,2]. WAT congregates in various sites, which may be generally categorized as subcutaneous adipose tissue (SAT) or visceral adipose tissue (VAT).

In addition to acting as a lipid storage depot, WAT is an active endocrine organ that secretes adipokines and adipocytokines to communicate with various other organs to influence systemic metabolism. Information regarding substrate availability, tissue mass, energy intake, and utilization is reported to the brain by adipose tissue [3,4]. The brain responds by making necessary changes to maintain the body homeostasis including energy balance. The satiety adipokine, leptin, is released in direct proportion to fat mass after a meal. Together with pancreatic insulin, leptin acts as an afferent signal to regulate the set point for fat stores [5,6,7], mediated by the hypothalamic-melanocortin pathway in the arcuate nucleus (ARC) [8] through regulation of the anorexigenic pro-opiomelanocortin (POMC), and the orexigenic neuropeptide Y (NPY)/agouti-related protein (AgRP) neurons. As another adipokine, adiponectin increases insulin sensitivity and fatty acid oxidation, suppresses hepatic glucose output, and inhibits inflammation [9]. Adipocytokines communicate with local and peripheral immune cells and can also influence metabolism. Interleukin-6 (IL-6), Interleukin 1-β (IL1-β), and tumor necrosis factor α (TNFα) increase local and systemic inflammation, recruit macrophages, and alter insulin signaling to promote insulin resistance [10,11,12].

Our lab and others have investigated a metabolic role for the aryl hydrocarbon receptor (AhR). Long studied for its importance in defending against environmental toxicants, AhR is a cell sensor and transcriptional regulator [13,14] with a range of physiological functions. Epidemiological studies demonstrate that AhR activation by persistent organic pollutants (POPs) promotes insulin resistance [15,16,17,18,19,20], although the mechanisms are not well understood. POPs, as well as certain dietary components, can be stored in adipose tissue [21] and act as ligands for AhR. Combined with a high-fat diet (HFD), the prototypical AhR ligand, 2,3,7,8—Tetrachlorodibenzo-p-dioxin (TCDD), can induce obesity in female mice [22]. The tryptophan (Trp) metabolite, kynurenine (Kyn), is a fat derivative that activates AhR to promote obesity [23]. Kyn is elevated in the circulation of obese subjects [24]. AhR activation by Kyn, TCDD, or other agonists inhibits adipogenesis and impedes lipolysis to reduce the release of free fatty acids (FFA), alters adipokine secretion, and promotes pro-inflammatory adipocytokine release [25,26,27,28]. Thus, a better understanding of the contribution of AhR within adipose tissue to systemic metabolism is critical to our understanding of obesity and metabolic illnesses.

The genetic depletion or pharmacological inhibition of AhR improves energy homeostasis in HFD-fed mice. Mice with a global depletion of (AhRKO), heterozygous AhR^+/−^ or a congenic low affinity AhR (B6.D2) are protected from HFD-induced obesity and metabolic dysfunction [29,30]. Moreover, AhR antagonists, such as with α-Napthoflavone or CH-223191, reverse the negative impact of HFD on systemic metabolism [31]. In comparison to AhRKO, tissue-specific depletion has yielded different and sometimes contradictory results. Specific gestational deletion of AhR from the liver exacerbates metabolic disease conditions, such as hepatic steatosis in animals fed HFDs, whereas conditional knockout from adult liver helps ameliorate this pathology [32,33].

To avoid the complications generated by the absence of AhR during critical developmental windows, inducible adipose-specific AhR depletion was used. Adipose-specific depletion from mature adipocytes protects from HFD-induced obesity and metabolic dysfunction in a sexually asymmetric manner. Females were robustly protected from diet-induced weight gain and glucose intolerance. Results in males were less profound, but suggestive of a delay in disease progression. These results highlight the possibility of AhR signaling in adipose tissue as a therapeutic target to combat obesity and insulin resistance in both the sexes.

## 2. Materials and Methods

### 2.1. Animals and Experimental Timeline

Experiments were approved by the Institutional Animal Care and Use Committee at Southern Illinois University School of Medicine in accordance with the Guide for the Care and Use of Laboratory Animals, as adopted and promulgated by the National Institutes of Health. AhR flox (AhR^fx/fx^ strain designation AhR^tm3.1Bra/J^) and Adiponectin-Cre with modified estrogen receptor (strain designation C57BL/6-Tg (Adipoq-iCre/ER^T2Soff/J^) were purchased from the Jackson Laboratory (Bar Harbor, ME). Adiponectin-CreER^T2^, linked to a Rosa^26loxP-STOP-loxP-tdTomato^ reporter, were crossed with AhR^fx/fx^ mice to generate Adiponectin-CreER^T2^::AhR^fx/fx^::Tomato^/+^ mice (Appendix A, left panel). These mice were administered two consecutive daily intraperitoneal (IP) injections of tamoxifen (T5648; Sigma-Aldrich (St. Louis, MO, USA)) at either 75 or 150 mg/kg to induce gene deletions (Appendix A, middle panel). Mice were rested for 1 week to allow for tamoxifen metabolism. PCR was used to analyze AhR^fx^ excision with forward primers 4062 (5′-GTCACTCAGCATTACACTTTCTA) and 4064 (5′-CAGTGGGAATAAGGCAAGAGTGA) in combination with the reverse primer 4088 (5′-GGTACAAGTGCACATGCCTGC), as previously described [34]. Male and female Adiponectin-CreER^T2^::AhR^fx/fx^::Tomato^/+^ with (CadKO) or without (WT) tamoxifen and global AhR null (AhRKO) mice were exposed to either a standard rodent chow (NCD, Lab diet formula 5001, Cat# 1319, St. Louis, MO, USA) or a 60% kcal high-fat diet (HFD, Research Diets #D12079B, #D12492, New Brunswick, NJ, USA) for 15 weeks. All mice were housed on corn cob bedding in a temperature-controlled facility. Body weight and food intake were assessed weekly. Extra precautions were taken to monitor food spillage. At week 13, animals were singly housed for indirect calorimetry studies (detailed below). During the subsequent week (week 14), activity behavior was monitored using infrared beam detectors (Minimitter, Bend, OR, USA). Glucose Tolerance tests (GTT) were performed at week 15. All animals were sacrificed by cervical dislocation to collect gonadal WAT (a VAT depot), SAT, BAT, hypothalamus, liver, muscle, and serum. All tissues were snap-frozen in liquid nitrogen. Some portions of liver, VAT, and SAT were fixed in 4% paraformaldehyde for 24 h, then processed for staining.

### 2.2. Activity Behavior Monitoring

Animals were housed in cages, within which the light chambers fitted with infra-red activity detectors (Minimitter, Bend, OR, USA). Actiview software was used to collect activity data into 6 min bins. Data were analyzed using Clocklab software (Actimetrics, Evanston, IL, USA).

### 2.3. Metabolic Chamber Measurements

Total and resting metabolic rates were measured via indirect calorimetry using the Oxy-max Indirect Calorimetry System/Metabolic Cage from Columbus Instruments (CLAMS, Columbus, OH, USA). Mice were individually housed in respiratory chambers. All comparisons were based on mice studied simultaneously in eight different respiratory chambers connected to the same infrared O_2_ and CO_2_ sensor to minimize the effects of environmental variation and instrument calibration. Mice were adapted to metabolic cages for 24 h before data collection. Gas samples were collected and analyzed every 5 min per animal; hourly averages were calculated. Output data from the software include O_2_ consumption (VO_2_) (ml kg^−1^ per minute), CO_2_ production (VCO_2_) (ml kg^−1^ per minute), respiratory quotient (RQ = VCO_2_/VO_2_), and heat production (heat = CV × VO_2_; CV = 3.815 + (1.232 × RQ)). Columbus Instruments Equations for Energy Expenditure provides details (http://www.colinst.com).

### 2.4. Glucose Tolerance Test (GTT)

After 15 h of fasting, a drop of blood from the snipped tail was used to test the glucose concentration using a Contour Next EZ Glucometer (Parsippany, NJ, USA), and mice were injected intraperitoneally with a bolus of 20% glucose at a dose 1 g/kg of body weight. Blood glucose was subsequently measured at 15, 30, 60, 90, and 120 min post injection.

### 2.5. Histological Analysis and Immunohistochemistry

Tissues were fixed in 4% paraformaldehyde, embedded in paraffin, sectioned, and stained with hematoxylin-eosin (H&E) and CD68 antibody at the Memorial Medical Center Histology Laboratory, Springfield IL. Quantification of adipocyte size was performed on H&E-stained sections using Image J software (version1.48e) Liver, VAT, and SAT were fixed in PBS + 10% formalin for 24 h, then stored in 70% ethanol. Single-blind, randomized images were obtained using a Nikon Eclipse E-600 microscopeequipped with an Olympus-750 video camera system. Adipocytes’ size was quantified using Image J Software (version1.48e) by an experimenter blinded to experimental conditions.

### 2.6. Immunofluorescence

Adipose and hypothalamus tissues were fixed in 4% paraformaldehyde for 24 h and then transferred into 20% sucrose in 0.1 M phosphate buffer for 24 h prior to sectioning. Furthermore, 20 µm adipose sections were collected serially using a cryostat (Model HM525 NX, ThermoFisher Scientific (Waltham, MA, USA). Sections were permeabilized with PBST (0.1 M PBS with 0.25% TritonX-100 (Fisher Scientific, Hampton, NH, USA)), washed, treated with sodium borohydride in PBS (1 mg/mL) for antigen retrieval, washed, blocked using 10%, normal goat serum/1% BSA for 1 h, and then incubated overnight in primary antibody (listed below) at 4 °C. Sections were then washed in PBST, incubated in secondary antibody (listed below and dilution) for 2 h, washed, and cover slipped using ProLong™ Gold (FisherScientific, Hampton, NH, USA) antifade reagent with DAPI. Images were captured using a Nikon Eclipse E-600 microscope equipped with an Olympus-750 video camera system. To include exposure time, filter selection, excitation intensity, contrast, and brightness, image settings were kept consistent across the groups. Staining intensities were measured using National Institute of Health Image J Software (Version 1.48e). Staining intensity was obtained after background staining was subtracted from mean intensities. Antibodies for immunofluorescence are listed in Appendix A.

### 2.7. q-PCR

Total RNA was extracted using Trizol (FisherScientific, Hampton, NH, USA), and cDNA was synthesized as previously described [29]. SYBR green-based real-time reverse transcriptase PCR was carried out on AB 1 step one plus real-time PCR system. Values for genes of interest were normalized using B_2_M as the housekeeping gene, and the relative levels of mRNA were determined using the ∆∆Ct method. Primer sequences for real-time PCR are listed in Appendix A.

### 2.8. Western Blot

VAT was homogenized in tissue protein extraction TPER lysis buffer (Fisher Scientific, Hampton, NH, USA) and then centrifuged at 12,000× *g* for 15 min at 4 °C. Protein was quantified using the BSA assay (Fisher Scientific, Hampton, NH, USA). Proteins were separated using SDS-page and transferred onto a nitrocellulose membrane. Membranes were blocked for 1 h with 5% serum BSA and incubated at 4 °C overnight with primary antibodies (Appendix A). Membranes were washed with TBST and incubated with secondary antibodies for 1 h at room temperature. Bands corresponding to the protein of interest were scanned using LI-COR Odyssey. Beta-actin was used as the loading control.

### 2.9. ELISA

Serum leptin (90030, Crystal Chem (Elk Grove Village, IL, USA)), insulin (90080, Crystal Chem (Elk Grove Village, IL, USA)), adiponectin (KMP0041, Novex/Invitrogen, Carlsbad, CA, USA)), and 17β-estradiol (ab108667, Abcam (Boston, MA, USA)) were measured by ELISA according to the manufacturer’s instructions.

### 2.10. Serum Triglyceride Assay

Triglycerides were quantified using a triglycerides liquid reagent set (Pointe Scientific INC. Cat# T7532-120 (Canton, MI, USA)).

### 2.11. Statistical Analysis

Data are presented as mean ± SEM. Rate of change regarding metabolic rate was calculated utilizing linear regression in GraphPad Prism Software (version 6.0). Where mentioned in the text, ANCOVA was utilized to determine if two slopes were different from one another. A one-way or two-way ANOVA with Tukey’s post hoc tests were utilized to identify significant differences between groups. *p* values of less than 0.05 were considered statistically significant.

## 3. Results

### 3.1. Conditional Deletion of AhR from Adipose Tissue

After the administration of tamoxifen to induce gene deletion (Appendix A, middle panel), tissues were processed and examined for the expression of the tdTomato reporter. High expression for tdTomato was observed in the tamoxifen-treated groups in adipose depots (visceral, subcutaneous, brown), suggesting high Cre recombinase activity (Figure 1a). A dose of 150 mg/kg tamoxifen increased the number of tdTomato-positive cells (Appendix A).

Following PCR, it was observed that primers [34] for the Ahr^fx^-excised allele (4062/4088) yielded a 180-bp band, whereas primers for the Ahr^fx^-unexcised allele (4064/4088) produced a 140-bp band (Figure 1b). The unexcised allele (140-bp) was present in all the tissues treated with tamoxifen in wild-type (WT) mice (Figure 1c). In contrast, tamoxifen treatment generated the excised allele (180-bp) in adipose tissue, demonstrating successful excision events (Figure 1c).

AhR functionality in a white adipose depot (VAT was used as a representative) in CadKO mice was determined by measuring the transcription levels of the downstream AhR gene (Cyp1b1) after treatment with the AhR agonist, β-naphthoflavone (BNF) (Appendix A, right panel). Cyp1b1 mRNA was similar to the vehicle-treated WT group in BNF-treated VAT from CadKO mice, whereas BNF produced a significant increase in Cyp1b1 mRNA in WT (50 mg/kg) (Figure 1d). Collectively, these data demonstrate tamoxifen-inducible Cre recombinase activity in adipose tissues and validate the depletion of AhR with compromised AhR functionality in CadKO mice. 

### 3.2. Sex-Specific Effects of Global and Adipose-Specific AhR Deletion on HFD-Induced Weight Gain and Metabolic Rate

Previous studies from our lab indicate that the global depletion of AhR (AhRKO) protects male mice from HFD-induced obesity and metabolic dysfunction [29,35]. This study compared diet-induced metabolic function in male and female AhRKO, CadKO, and WT mice (Figure 2a). Weight gain within sex was not different among genotypes on NCD (Figure 2b,c). On HFD, AhRKO, and CadKO, males gained less weight than WT, but more weight than all genotypes on NCD, with significant differences apparent at week 4 (Figure 2b). Interestingly, the statistical significance between WT and CadKO was reduced at week 11 and was no longer seen at week 12. In contrast, AhRKO maintained a robust significant difference from WT throughout the experiment. AhR depletion in females completely negated HFD-induced weight gain (Figure 2c). HFD-fed CadKO and AhRKO females weighed less than WT as early as week 2, and differences were maintained throughout the experiment. HFD-fed female AhRKO and CadKO weighed the same as the NCD-fed mice of all genotypes (Figure 2c).

Net calorie intake was similar in males of all genotypes on both the diets (Figure 2d, Appendix A). However, both HFD-fed CadKO and AhRKO females consumed a similar amount of calories as the NCD-fed females of all genotypes, which was significantly lower than HFD-fed WT females (Figure 2d, Appendix A). Indirect calorimetry suggested that energy expenditure (EE) was not different among all male genotypes on HFD and was solely related to mass, indicated by overlapping regression lines with slopes that were not significantly different (Figure 2e). Similarly, daily overall VO_2_, VCO_2_, and mass-dependent EE were unchanged in CadKO and AhRKO males (Appendix A). In contrast, both AhRKO and CadKO females displayed higher mass independent EE on HFD, indicated by regression lines with significantly different slopes compared to WT (Figure 2f). Daily overall VO_2_, VCO_2_, and mass dependent EE were also significantly higher in CadKO and AhRKO females (Appendix A). The calorimetry data for females collectively indicates higher metabolic activity in females when AhR has been depleted. The respiratory quotient was in the range of 0.7–0.8 among all genotypes, suggesting that fat was used as the primary fuel source among those on a HFD (Appendix A).

Locomotor activity was examined to investigate the influence of physical activity on energy output. WT mice displayed less overall activity on HFD compared to NCD for both males and females (Appendix A). However, activity of AhRKO and CadKO animals on HFD was maintained at levels similar to NCD. No sex differences in activity were observed. Overall, these data suggest that AhRKO and CadKO protects mice from HFD-induced weight gain, with a greater effect being identifiable among females.

### 3.3. CadKO Improves Systemic Glucose Homeostasis and Protects against HFD-Induced Insulin Resistance in Females

To investigate whether protection from weight gain in CadKO is concurrent with improved glucose homeostasis, glucose tolerance (GTT) was examined. Neither sex nor genotype affected fasting glucose on NCD (Figure 3a). HFD increased fasting glucose in WT males and females. As expected [29], AhRKO males had decreased fasting glucose on HFD compared to WT. However, fasting glucose in HFD-fed CadKO males was similar to WT. In contrast, both the AhRKO and CadKO females exhibited significantly lower fasting glucose on HFD than WT (Figure 3a).

AhRKO and CadKO males showed significantly better glucose tolerance than WT on NCD, as evidenced by a lower area under the curve (AUC) on GTT (Figure 3b,d). Glucose tolerance in HFD-fed males was not different among genotypes, although *p* = 0.07 was demonstrated for AhRKO (Figure 3 3e,g). GTT was not different among female genotypes on NCD (Figure 3c,d); however, both AhRKO and CadKO females displayed significantly better glucose tolerance on HFD compared to WT on HFD (Figure 3f,g).

As glucose homeostasis strongly correlates with insulin secretion, we measured serum insulin levels. Insulin levels in WT males were substantially increased by HFD compared to CadKO, suggesting HFD-induced hyperinsulinemia (Table 1), similar to previous results in AhRKO males [29]. In contrast, insulin was unchanged by HFD in both WT and CadKO females (Table 1). Collectively, these data suggest that the depletion of AhR in adipose tissue alone does not afford the same level of protection as global AhR depletion under HFD in males. However, CadKO males are protected from HFD-induced hyperinsulinemia, suggesting reduced stress on pancreatic β-cells. In contrast, the depletion of AhR from adipose tissue alone in females is sufficient to provide the same widespread protection from HFD-induced changes in glucose metabolism, as is seen in global AhRKO mice.

### 3.4. CadKO Protects from HFD-Induced Adiposity in Females and Inflammation in Males

In males, gross morphology (Appendix A) and overall fat mass (Figure 4a) suggests CadKO did not completely protect from overall accumulation of body fat with HFD. VAT weight was similar for both WT and CadKO males on HFD (Figure 4b). However, CadKO males were protected from increased SAT mass on HFD (Figure 4b). As males tend to accumulate fat in VAT before SAT, this may suggest that VAT was saturated, while SAT was not. In females, gross morphology (Appendix A) and overall fat mass (Figure 4a) was significantly reduced in HFD-fed CadKO females (~5% fat mass) compared to HFD-fed WT females (~13% fat mass). VAT and SAT were significantly higher in WT females (VAT = 8.5%, SAT = 4.6%) compared to CadKO females (VAT = 3.5%, SAT = 2.9%) (Figure 4b).

The morphology of adipose depots was examined with H&E staining in VAT and SAT. HFD-fed WT and CadKO males increased adipocyte size with a broad distribution of adipocyte sizes (Figure 4c,d; Appendix A). CadKO females had an increased percentage of smaller adipocytes with HFD, similar to NCD-fed females (Figure 4e,f; Appendix A). In contrast, WT HFD-fed females had a broad distribution of adipocyte sizes, similar to HFD-fed WT and CadKO males (Figure 4e,f; Appendix A).

H&E staining of VAT in HFD-fed WT males displayed the presence of crown-like structures (CLS) that were absent in all other groups (Figure 4c,e). CLS represent mononuclear cell infiltration around an unhealthy adipocyte. CD68 (a monocyte/macrophage marker) staining highlighted the presence of cytoplasmic macrophages around adipocytes in WT males on HFD, which were not observed in CadKO HFD-fed males (Figure 4c, inserts). CLS around dead adipocytes and macrophage infiltration is typical in obese, metabolically unhealthy individuals. CLS were not observed within SAT in any of the groups (Appendix A). Since changes in VAT are associated with poor metabolic health, our subsequent analysis of WAT focused mainly on VAT.

Because CLS in VAT is congruent with inflammation, inflammatory pathways were explored in VAT. Transcript levels were expressed as a ratio of HFD/NCD to accentuate changes associated with diet (value > 1 represents increase in HFD compared to NCD). TNFα transcripts were increased by HFD in WT males (HFD/NCD = 3.1) but were not changed by HFD in CadKO males (HFD/NCD = 1.4) (Figure 4g). Similarly, IL1β transcripts were increased by HFD in WT males (HFD/NCD = 5.5) compared to CadKO males (HFD/NCD = 1.1) (Figure 4g). In contrast, HFD did not significantly elevate TNFα (HFD/NCD = 1.4) or IL1β (HFD/NCD = 1.8) in WT and CadKO females (Figure 4g).

The AhR/Stat3/IL6 pathway was investigated as the activation of AhR by fat derivatives within the HFD, e.g., by kynurenine (Kyn), has been reported to help this pathway induce an obese phenotype [36]. We observed AhR/Stat3/IL6 pathway attenuation in VAT for CadKO mice of both sexes (Figure 4h), revealing another pathway by which AhR depletion may improve metabolic function. Other adipose tissue metabolic pathways were also examined in VAT. The transcript levels of PPARγ, an indicator of adipogenesis, were not altered by HFD in WT (HFD/NCD = 1.6) or CadKO (HFD/NCD = 1.2) males (Figure 4i) or in WT HFD-fed females (HFD/NCD = 1.7) (Figure 4i). In contrast, female CadKO mice had significantly increased PPARγ transcripts on HFD (HFD/NCD = 4.7). Lipolysis in WAT was investigated by examining Hsl, its rate-limiting enzyme. Hsl transcripts were reduced by HFD in WT (HFD/NCD = 0.4) and CadKO (HFD/NCD = 0.5) males (Figure 4i). In contrast, HFD increased Hsl expression in both WT (HFD/NCD = 1.9) and CadKO (HFD/NCD = 2.7) females (Figure 4i). The differences in Hsl transcripts prompted the investigation of serum triglycerides. In males, HFD increased serum triglycerides (CadKO = 195.88 mg/dL; WT = 181.74 mg/dL) in both the genotypes compared to NCD (50–150 mg/dL) (Table 1). Similarly, HFD also increased triglycerides in females (CadKO = 155.33 mg/dL; WT = 173.2 mg/dL) compared to NCD (40–120 mg/dL) (Table 1). However, the level in CadKO females was lower than that of WT females and lower compared to the males (Table 1). The comparable serum triglyceride levels observed in females, despite their lower high-fat diet intake and body weight, along with the increased expression of HSL transcripts, suggest that CadKO females may exhibit enhanced fat utilization and mobilization compared to WT females.

Lipogenesis is also associated with HFD-induced metabolic diseases. Levels of Srebp1c, a transcription factor that regulates genes associated with lipogenesis, were not changed by HFD in CadKO males (HFD/NCD = 1.3), although they were reduced by HFD in male WT (HFD/NCD = 0.5) (Figure 4i). However, in female mice, Srebp1c transcripts were reduced by HFD in both WT (HFD/NCD = 0.2) and CadKO (HFD/NCD = 0.4) mice (Figure 4i). To investigate the sexual dichotomy, we measured ERα expression, as activated AhR attenuates ERα expression. ERα expression was reduced by HFD in males for both WT (HFD/NCD = 0.4) and CadKO (HFD/NCD = 0.6) (Figure 4i). In females, HFD reduced ERα in WT (HFD/NCD = 0.6), but not in CadKO (HFD/NCD = 1.1) (Figure 4i), suggesting ERα signaling may mediate the differences between sexes. Overall, these data suggest that the adipose-specific depletion of AhR provides a mechanism to reduce adiposity in female mice and to reduce inflammation in male mice.

### 3.5. Adipose-Specific Deletion of AhR Protects Endocrine Function of White Adipocytes in Females

Circulating leptin provides information about the amount of fat stores and possible leptin resistance. Serum leptin was not different across sex or genotype on NCD but was significantly elevated by HFD in all groups (Table 1). No significant differences occurred among male groups on HFD. In females, leptin levels in WT HFD-fed animals increased almost to the level in males. In CadKO HFD-fed females, the leptin levels were elevated but significantly less so than in WT HFD-fed females (Table 1), suggesting potential protection from HFD-induced leptin resistance.

Adiponectin reduces inflammation and improves insulin sensitivity and is typically higher in individuals with metabolically healthy body fat but lower in individuals that are obese or have unhealthy body fat. On NCD, males demonstrated significantly lower adiponectin compared to females (Table 1). CadKO males on NCD had higher adiponectin levels than WT mice on NCD (Table 1), which is consistent with GTT studies (Figure 3b,d), suggesting improved insulin sensitivity in CadKO vs. WT males on NCD. Males on HFD for both genotypes increased fat mass (Figure 4a; Appendix A) but failed to increase serum adiponectin to the level of HFD-fed WT females (Table 1), suggesting an unhealthy metabolic condition. In HFD-fed females, adiponectin was significantly higher in WT females compared to CadKO females (Table 1) despite increased fat mass in HFD-fed WT females (Figure 4a; Appendix A), suggesting that both female groups remained metabolically healthy. When adiponectin was normalized to fat mass, CadKO females on HFD demonstrated considerably higher adiponectin levels (39.1 mg/mL/g) compared to the other groups (WT female = 17.8, WT male = 18.5; CadKO male = 17 mg/mL/g) (Appendix A). Similarly, adiponectin transcript levels were elevated in CadKO females on HFD (HFD/NCD = 2.9) compared to WT females (HFD/NCD = 0.9), WT males (HFD/NCD = 1.4), and CadKO males (HFD/NCD = 1.2) (Appendix A). Overall, these data suggest AhR specific deletion from adipose tissue attenuates leptin and elevates adiponectin in females to maintain healthy fat and protect against diet-induced metabolic dysfunction.

### 3.6. HFD-Fed Female CadKO Mice Have Increased Leptin and Estrogen Receptors

Because CadKO females consume fewer calories on HFD compared to WT females (Figure 2d, Appendix A), changes in leptin and estrogen signaling in the hypothalamic feeding center were examined. Lower serum leptin levels in female HFD-fed CadKO mice suggests a possible protection from HFD-induced leptin resistance (Table 1). To examine the possibility of leptin resistance, leptin receptor (LepR) expression in the arcuate (ARC), nucleus, and ventromedial (VMH) nucleus of the hypothalamus were investigated. As no difference in leptin levels were observed in the NCD groups (Table 1), LepR levels were examined only in the HFD groups. WT and CadKO males, as well as WT females, had significantly lower LepR in ARC and VMH nuclei compared to CadKO females (Figure 5b,c). Next, serum 17β-estradiol and ERα expression in the hypothalamic feeding center were examined to investigate estrogen signaling. Serum 17β-estradiol was not affected by diet, genotype, or sex (Table 1). ERα (Figure 5a) was significantly higher in CadKO females compared to all other HFD groups (Figure 5d,e). Overall, these data suggest CadKO protects females from leptin resistance and improves estrogen signaling.

To investigate whether differences in leptin and estrogen signaling affects thermogenesis, non-shivering thermogenesis in BAT was explored. HFD had no effect on transcript or protein levels of the thermogenic gene, uncoupling protein 1 (UCP1), which conducts protons across the inner mitochondrial membrane to facilitate the uncoupling of oxidative phosphorylation from ATP synthesis and dissipate heat (Appendix A). Body temperature values were also not different in any of the groups (Appendix A), suggesting CadKO had no effect on body temperature or in BAT thermogenesis. During cold exposure, the liver releases Fgf21 to activate the sympathetic nervous system, inducing BAT thermogenesis. Similar to UCP1, Fgf21 transcript levels in the liver on HFD were not different between the sexes of the CadKO and WT groups (Appendix A). Unaltered liver Fgf21 and BAT UCP1 expression (Appendix A) suggests that the Fgf21-induced BAT thermogenesis pathway in the liver was unaffected by CadKO in either sex. Overall, these data suggest that AhR-specific deletion from adipose tissue in female improves leptin and estrogen signaling in energy regulatory regions under HFD to negate HFD-induced weight gain via reduced calorie intake and an increase in EE, which is not mediated by BAT thermogenesis.

### 3.7. CadKO Protection against Liver Pathogenesis Is Sexually Dimorphic

To examine whether alterations in adipose biology can exert a systemic effect, we examined liver tissue as it plays a central role in both fatty acid and glucose metabolism. HFD increased liver mass in WT males but not in CadKO males (Appendix A). Liver weight was not significantly different in HFD-fed females of either genotype (Appendix A). H&E staining of the liver revealed a normal hepatocellular architecture in the NCD groups (Figure 6a). In contrast, large cytoplasmic spaces and intracytoplasmic vacuoles, were observed in HFD-fed WT animals, consistent with lipid accumulation, with males showing more histopathology compared to females (Figure 6b). Liver histology was improved in HFD-fed CadKO males and females, with CadKO females being indistinguishable from NCD groups (Figure 6b; Appendix A).

Several genes controlling fatty acid and glucose metabolism in the liver were altered by HFD in a sex-specific way. Pparα (a gene involved in regulating fatty acid oxidation and energy homeostasis) was elevated by HFD in WT (HFD/NCD = 2.1) and CadKO (HFD/NCD = 3.2) males (Figure 6c). However, Pparα in female WT was not changed by diet (HFD/NCD = 0.9), and HFD led to a reduction in Pparα in female CadKO mice (HFD/NCD = 0.5) (Figure 6c). Transcripts for the fatty acid translocase, CD36 (which assists in the uptake of fatty acids), increased slightly on HFD for male WT (HFD/NCD = 1.4), male CadKO (HFD/NCD = 1.2), and female WT mice (HFD/NCD = 1.3) (Figure 6c). In contrast, CadKO female mice on HFD significantly suppressed CD36 level (HFD/NCD = 0.5) (Figure 6c). Transcripts for the lipogenic gene, ACC (acetyl Coenzyme A carboxylase) (catalyzes the final step of fatty acid biosynthesis), were not altered by HFD in male WT (HFD/NCD = 1.1) and CadKO mice (HFD/NCD = 1.2) (Figure 6c). However, HFD significantly increased ACC expression in WT females (HFD/NCD = 2.1) compared to CadKO females (HFD/NCD = 1.2) (Figure 6c). Finally, transcripts of glucose 6 phosphatase (G6Pase) (catalyzes the final steps of both gluconeogenesis and glycogenolysis) were examined. HFD had no effect on G6Pase levels in WT males (HFD/NCD = 1.1) but G6Pase was significantly elevated in HFD-fed CadKO males (HFD/NCD = 2.8) (Figure 6c). G6Pase was not changed by HFD in WT (HFD/NCD = 0.8) or in CadKO (HFD/NCD = 0.9) females (Figure 6c). Overall, these data suggest that the protective effects of CadKO on HFD-induced hepatic pathology may be mediated by different mechanisms for males (more fatty acid metabolism) and females (impaired entry and synthesis of fat).

## 4. Discussion

This study shows that males and females regulate systemic metabolic processes differently, especially when it comes to fat and glucose metabolism, which may lead to sex-based differences in metabolic diseases. Evolutionarily, sex-dependent variations in survival strategies suggest that males prepare for periods of energy absence by increasing food/energy intake to increase fat stores, whereas females react to energy scarcity to survive by decreasing EE and preserving fat stores [37]. These observations are congruent with the female storage of fat in SAT (long-term storage) and male utilization of VAT (metabolically active) [38,39]. These sex-specific differences also manifest in disease pathology. While females are more prone to obesity, males are more susceptible to obesity-related comorbidities, such as type 2 diabetes [4,40].

To avoid any effects of tamoxifen on the study outcome, a 7-day wash out time was provided before initiating feeding. After five consecutive daily doses (100 mg/kg/day), only 1% remained in WAT 10 days after the last gavage [41]. Furthermore, our study examined the effects of a HFD over a period of 12 weeks after tamoxifen treatment. The CadKO model demonstrates that AhR is an important regulator of adult adipose tissue (Figure 4a–i; Appendix A), mediating the systemic regulation of body weight to a greater extent in females than in males (Figure 2b,c). Other investigators have reported similar findings in male mice after depleting AhR function by inhibition [31,42], reduced AhR affinity (C56BL/6.D2) [30,42], whole body knock out [29], or inducible liver-specific knock out [33]. Collectively, AhR depletion protects against weight gain from HFD in male mice. The few studies that examined females also revealed protection from HFD to a greater extent compared to males [33,42]. In contrast, one previous study [43] used germline AhR depletion from adipocytes and unexpectedly found an increase in body weight in males on HFD. Although the current study’s results failed to demonstrate robust protection from HFD-induced weight gain in males (Figure 2b), weight gain was certainly not exacerbated. However, adipose-specific AhR depletion using platelet-derived growth factor receptor alpha (Pdgfrα)-Cre mice, where AhR is depleted beginning at the preadipocyte phase, demonstrated reduced weight gain in males, similar to this study [44]. Discrepancies could reflect the timing of AhR deletion. The inducible depletion of preadipocytes or mature adipocytes allows for AhR to be present during development and affects only adult adipose tissue biology, whereas germline depletion occurs throughout the lifespan.

Over time, changes in body weight and fat mass can develop with very small mismatches in food intake and energy expenditure [45]. AhR deficiency in females led to reduced calorie intake (Figure 2d) and increased metabolic rate to protect from HFD-induced weight gain (Figure 2f). In males, most previous studies showed no change in consumed calories on HFD [29,30,43], although a low-affinity AhR mice strain reduced food consumption [46]. Furthermore, global AhR-deficient male mice (AhRKO) maintain a lean phenotype on HFD due to an increase in EE [29]. In this study, male CadKO mice did not show increased EE on HFD (Figure 2e). However, both female CadKO and AhRKO mice did increase EE under HFD conditions (Figure 2f), similar to a report from an inducible liver-specific AhR knock out [33]. Collectively, the available data demonstrate that AhR deficiency in mature adipose tissue has sex-specific effects on EE. Also, CadKO males showed increased activity, likely driven by testosterone [47], which may protect from weight gain on HFD (Figure 2b).

This study and others [44] suggest that males require AhR deficiency beyond adipose tissue to regulate glucose homeostasis. Males were not protected from developing glucose insensitivity on HFD (Figure 3a–g). Tissue-specific AhR depletion at the preadipocyte phase in males (Pdgfrα-Cre) produced comparable effects on glucose tolerance [44], while the adiponectin-Cre model (starting at fertilization) showed impaired glucose homeostasis in HFD-fed males [43]. Hyperinsulinemia in WT males on HFD (Table 1) suggests a developing insulin resistance with pancreatic β-cell compensation to maintain glucose levels. Both AhRKO and AhR^+/−^ male mice were protected from hyperinsulinemia on HFD [29]. CadKO females show improved fasting glucose and glucose tolerance, similar to AhRKO (Figure 3a–g), indicating that adipose AhR may play a bigger role in mediating systemic glucose sensitivity in females (Figure 3a–g), which supports a role for AhR in the regulation of insulin levels [16,48,49].

AhR depletion from adipose tissue clearly has sex-specific effects; females maintain small adipocytes on HFD (Figure 4e,f; Appendix A). Morphological and transcript-level analysis shows the maintenance of small adipocytes in CadKO females on HFD (Figure 4; Appendix A), consistent with healthy adipose function with active adipogenesis and lipolysis, which was not seen in males. Similarly, Pdgfrα-Cre Ahr-floxed (Ahr^fl/fl^) knockout mice demonstrated protection from HFD-induced obesity and maintained small adipocyte size [44]. Similarly, Pdgfrα-Cre Ahr-floxed (Ahr^fl/fl^) knockout mice demonstrated protection from HFD-induced obesity and maintained small adipocyte size [44]. Small adipocytes store and release fat more efficiently. The proliferation of small adipocytes (hyperplasia) can result in adiposity with maintenance of metabolic function, a metabolically healthy, obese state. Adipocyte hypertrophy leads to dysfunction and an inability to remove lipid from the blood. On the other hand, similar to effects on weight and glucose sensitivity, preadipocyte-specific AhR depletion [44] showed exacerbated HFD-induced obesity with larger adipocytes in males. The authors speculated that the unexpected HFD-induced obesity with larger adipocytes could be due to enhanced triglyceride synthesis/de novo lipogenesis. Without AhR, there is potential to upregulate de novo lipogenesis, as lipogenesis is inhibited by AhR activation [50]. Although the male CadKO mice in this study were protected from HFD-induced weight gain, it is possible that de novo lipogenesis increased, causing elevated adiposity and reduced lean body mass. Additionally, AhR knockout in mature adipocytes vs. in developing preadipocytes might allow for more effective communication between preadipocytes and mature adipocytes regarding the process of adipogenesis. The delay of knockout of AhR until the adipogenesis program is fully activated and adiponectin is expressed may produce a completely different phenotype in response to AhR ligands.

AhR activation in mature adipocytes, perhaps in response to ligands from HFD, inhibits adipogenesis and lipolysis through PPARγ and HSL pathways, respectively [25,26,28,30,51]. CadKO females increased PPARγ in response to HFD (Figure 4i), which promotes adipogenesis and may explain the maintenance of smaller adipocytes in female CadKO. The upregulation of HSL (Figure 4i) in females indicates enhanced lipolysis, which increases the availability of FFA for energy during fasting and exercise. The rate-limiting lipogenic gene, Srebp1c, was significantly higher in male CadKO VAT (Figure 4i). Male HFD-fed WT mice have low Srebp1c levels in VAT (Figure 4i), which may reflect inflammation and insulin resistance. Sex-specific differences were apparent in the expression of this transcript, which was not surprising, as males are known to increase de novo lipogenesis in VAT when presented with excess calories, explaining why VAT mass was similar in CadKO and WT males (Figure 4b). HFD-induced lipogenesis can be beneficial for reducing ectopic lipids in adjacent organs.

Although the mechanism underlying sex differences remains unclear, the possibility of AhR interactions with the female sex steroid receptor, Erα, were explored. AhR inhibits the signaling pathways of ERα by recruiting ERα away from estrogen responsive elements in target genes, increasing the proteasomal degradation of ERα and increasing the synthesis of an inhibitory protein [52]. In females, ERα inhibits WAT development, amount, and size, suggesting ERα also regulates triglyceride accumulation [53,54]. In this study, deleting AhR from adipose tissue increased the expression of ERα in female VAT. An increase in ERα favors the anti-obesity phenotype in CadKO females on HFD.

In males, AhR depletion in adipose tissue may improve crosstalk with local and peripheral immune cells, as suggested by inflammatory markers (Figure 4g,h). HFD-fed CadKO males showed reduced proinflammatory adipocytokines (TNFα, IL1β, IL6), thus reducing inflammation, macrophage infiltration, and hyperinsulinemia (Figure 4c; Table 1). AhR activation by obesogenic POPs, such as coplanar PCBs and TCDD, increased the inflammation of murine and human adipocytes; selective ablation of AhR in adipose tissue abolishes the negative effects on adipose tissue inflammation [27,55,56,57]. Moreover, AhR activation by obesogens and other metabolites present in HFD, such as Kyn, also negatively affects adipose function through the AhR/Stat3/IL6 pathway [36].

Glucose sensitivity was not different from WT in HFD-fed CadKO males (Figure 3a,e,g). In contrast, global AhR depletion preserves glucose homeostasis in HFD-fed males. HFD-fed AhRKO males showed reduced adipocytokines, increased adiponectin secretion, and improved leptin signaling [29]. This might contribute to the preservation of healthy fasting and postprandial glucose levels. Adipose-specific depletion in male mice also protected adipose tissue from HFD-induced inflammation and macrophage infiltration [43]. The same study also revealed exacerbated weight gain on HFD in male CKO mice, highlighting that AhR depletion from adipose tissue is more effective in protecting adipose tissue from inflammation than in protecting weight homeostasis.

CadKO females consumed fewer calories on HFD (Figure 2d). Increased ERα levels in hypothalamic regions of CadKO female (Figure 5d,e) may explain the sex-specific differences. Hypothalamic brain regions associated with food intake and energy metabolism, including the arcuate nucleus (ARC) and ventromedial nucleus of the hypothalamus (VMH), express high levels of ERα [58], and the injection of 17β-estradiol can decrease food intake [59,60,61]. Adipose–hypothalamus cross talk, with positive signals generated from healthy, lean adipocytes in CadKO females (Figure 4a,e), may provide a mechanism for adipose-specific AhR deletion, affecting hypothalamic ERα. Adipose-derived leptin signals are sent to the hypothalamus, which in turn, alters adipose tissue dynamics. Moreover, adiponectin can also promote fatty acid oxidation and improve insulin sensitivity mediated by β-oxidation and AMP-activated protein kinase (AMPK) [62,63,64]. Although β-oxidation was not explored in this study, increased serum triglycerides, adiponectin, and lipolysis gene transcripts indicate that the observed increase in EE in HFD-fed CadKO females may involve an improved use of FFA. Typically, adult females have both increased leptin levels and sensitivity compared to males [65,66,67]. Healthy adipose–hypothalamic crosstalk and better leptin signaling may provide important mechanisms to protect CadKO females from HFD-induced weight gain.

Males may be more susceptible to HFD-induced liver damage (Figure 6a,b). Communication between adipose tissue and the liver is a significant contributor to changes in hepatocyte biology leading to fatty liver. Enlarged, metabolically unhealthy VAT seen in HFD-fed WT males (Figure 4c) leads to systemic insulin resistance, hyperinsulinemia, and inflammation initiated by inflammatory adipocytokine release and subsequent energy influx into the liver, which enhances vulnerability to metabolic stress [68]. Furthermore, hyperinsulinemia induces the deposition of liver fat [69], consistent with the histological findings in WT male livers (Table 1 and Figure 6b), although lipid droplets in the liver were not directly assessed. Protection in CadKO males was apparent at the molecular level in the liver. PPARα, which is important for mitochondrial fatty acid β-oxidation, was upregulated in CadKO males (Figure 6c). However, PPARα expression in the HFD liver did not increase in AhRKO and AhR^+/−^ males [29,35]. As AhR is still available in the liver of CadKO mice, increased PPARα may be an adaptive response to attenuate fat accumulation in the liver [70]. The liver also plays a vital role in glucose homeostasis. Gluconeogenesis and glycogenolysis help maintain normal blood glucose. The activation of AhR by obesogens can suppress gluconeogenesis and glycogenolysis, which is attenuated in the low-affinity AhR mouse strain, C57BL/6.D2 [71], which are less susceptible to the negative metabolic impacts of HFD. In this study, male CadKO mice on HFD experience a marked increase in G6Pase transcripts, the rate-limiting enzyme for both gluconeogenesis and glycogenolysis (Figure 6c). Therefore, in males, the absence of AhR from adipose promotes crosstalk with the liver metabolic pathways, which results in an increased generation of glucose. Molecular analyses demonstrated that CadKO females and males utilize unique strategies for metabolic protection. The anti-inflammatory adipokine, adiponectin, which is higher in females, is protective against fatty liver [72]. Sex-specific differences in adiponectin levels, the appearance of fatty liver, and molecular changes were apparent (Appendix A, Figure 6b). CD36, an AhR target gene [73] that promotes FFA translocation into the liver, was reduced in HFD-fed CadKO females (Figure 6c), suggesting a mechanism for protection from hepatic steatosis (Figure 6a,b). Hepatic lipogenesis was reduced in female CadKO, as evidenced by decreased ACC transcripts (Figure 6c), also promoting protection from steatosis. Additionally, increased adipogenesis capacity in female CadKO WAT may allow for the deposition of additional lipids in healthy fat and prevent lipid spillover in other tissues.

The deletion of AhR from adipose tissue provides a good working system to examine the effects of HFD-induced metabolic dysfunction in a sexually dimorphic way (Figure 7). Improved adipose biology in CadKO mice is likely the cause of increased weight gain and insulin resistance on HFD for both sexes, although specific mechanisms are sexually dimorphic. Local effects within adipose tissue and interactions with the immune system may be more important for males. This might help to restrict adipose tissue inflammation, which was observed in WT males. Protection from adipose tissue inflammation was reflected by lower levels of proinflammatory adipocytokines, reduced macrophage infiltration, and the absence of CLS. This protection helps maintain normal adipose physiology in CadKO males, as indicated by less ectopic lipid spillover in the liver and protection from insulin resistance. In contrast, female CadKO mice on HFD are protected from weight gain and hyperglycemia at least in part through decreased food intake and elevated EE. Their adipocytes remain small and healthy through increased adipogenesis and lipolysis. The maintenance of small, healthy adipocytes facilitates the secretion of beneficial adipokines, such as adiponectin and leptin, and can promote healthy crosstalk with other metabolic organs. Healthy communication with feeding centers of the hypothalamus may help to maintain healthy energy balance and prevent metabolic dysfunction. Importantly, this study reveals the significance of studying both sexes, and highlights the potential need for developing therapies based on sex. Since females are prone to obesity etiology and males to diabetes, understanding sex-specific pathophysiological changes may lead to new therapeutic targets, including AhR and its downstream signaling pathways.

## Figures and Tables

**Figure 1 cells-12-01748-f001:**
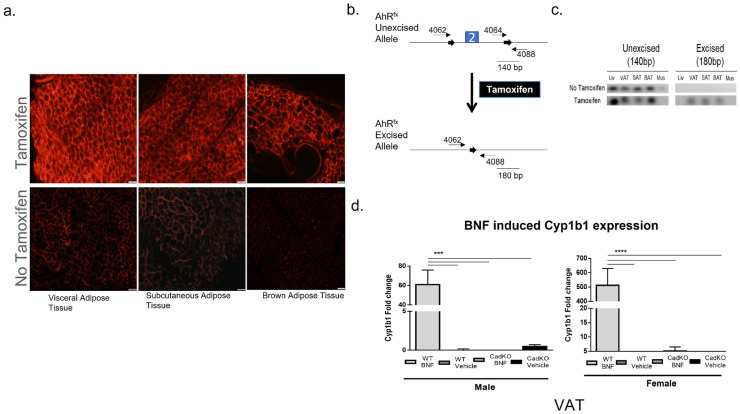
Specificity of Cre recombinase-mediated excision of AhR^fx^ allele upon tamoxifen administration. (**a**) Tomato expression with or without tamoxifen in adipose depots. Upper panel: tomato expression in visceral (VAT), subcutaneous (SAT), and brown (BAT) adipose tissue cluster (from left to right) with 150 mg/kg tamoxifen. Lower panel: tomato expression in VAT, SAT, and BAT cluster (from left to right) without tamoxifen. Scale bar = 50 μM (**b**) Diagrammatic representation of the Ahr^fx^-unexcised and the Ahr^fx^-excised alleles. Solid lines represent the fragment sizes generated by PCR amplification of the Ahr^fx^-unexcised and Ahr^fx^-excised allele using the forward primers (4062 and 4064) and the reverse primer (4088). 2 represents Exon 2 in the AhR gene. (**c**) Specificity of excised events was determined by genotyping for both the unexcised (left panel) and excised (right panel) alleles in genomic DNA of various tissues collected from Adiponectin^CreERT2^:AhR^fx^ mice with (upper panel) or without (lower panel) tamoxifen. Liv: Liver; VAT: Visceral Adipose Tissue; SAT: Subcutaneous Adipose Tissue; BAT: Brown Adipose Tissue; Mus: Muscle. (**d**) VAT were harvested, and total RNA was extracted to quantify mRNA expression level via real-time PCR analysis. The level of Cyp1b1 (downstream of AhR gene) mRNA expression in VAT upon BNF or Vehicle treatment in male (left) and female (right) CadKO and WT mice. PCR data were normalized against the amount of B2M. n = 4–5 for each group by one-way ANOVA with Tukey’s post hoc comparison. *** *p* < 0.001, **** *p* < 0.0001 by one-way ANOVA with Tukey’s post hoc comparison.

**Figure 2 cells-12-01748-f002:**
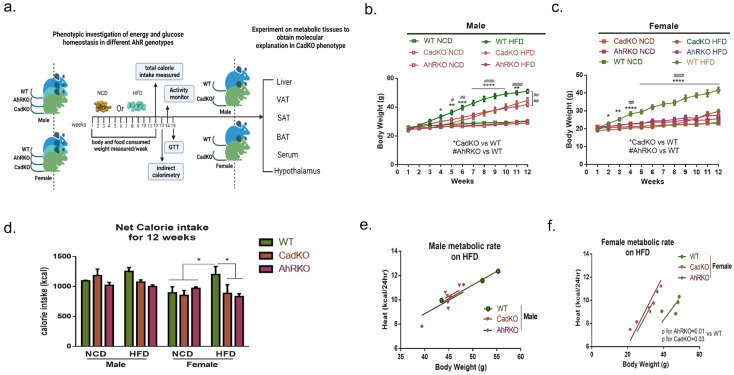
Global and adipose-specific AhR deletion protect mice from HFD-induced weight gain in sexually asymmetric mechanism. (**a**) Schematic representation of experimental design to examine phenotypical comparison between different AhR genotypes. Created by BioRender.com. (**b**,**c**) Body weight gain in AhRKO (n = 8), CadKO (n = 8), and WT (n = 8) mice for 12 weeks of NCD and HFD in males (**b**) and females (**c**). (**d**) Total average kilocalories consumed by AhRKO (n = 4), CadKO (n = 4), and WT (n = 4) mice after 12 weeks of NCD (3.36 kcal/g) and HFD (5.21 kcal/g). (**e**,**f**) Total mass-independent metabolic rates of AhRKO, CadKO, and WT mice were measured at week 13 of HFD by using the Oxymax Indirect Calorimetry System/Metabolic Cage from Columbus Instruments (CLAMS) for 24 h. * represent CadKO vs. WT and ^#^ represent AhRKO vs. WT on HFD (**b**,**c**). Two-independent study (n = 4, for each group). *^,#^ *p* < 0.05, **^,##^
*p* < 0.01, *** *p* < 0.001, ****^,####^
*p* < 0.0001 by 2-way ANOVA with Tukey’s post hoc comparison. ANCOVA was utilized to determine if two slopes were different from one another (**e**,**f**).

**Figure 3 cells-12-01748-f003:**
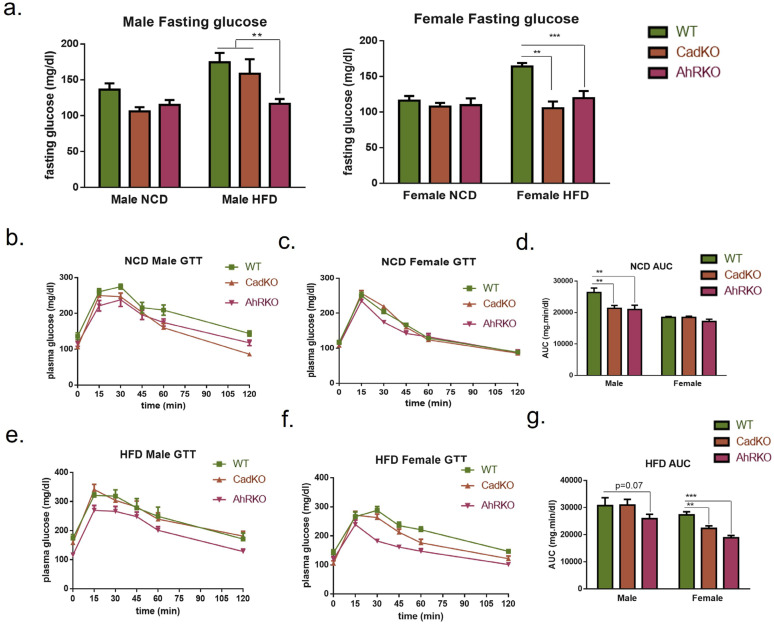
CadKO improves fasting glucose and systemic glucose tolerance in females on HFD. (**a**) Fasted male and female blood glucose levels (fasting for 15 h) were obtained from tail vein of WT (n = 8–10), CadKO (n = 8–10) and AhRKO, (n = 8–10) mice after 15 weeks of NCD and HFD feeding. Two-independent study (n = 4–5, for each group). (**b**–**g**) GTT after 15 weeks of NCD (**b**,**c**) and HFD feeding (**e**,**f**) for male (**b**,**e**) and female (**c**,**f**) WT, CadKO, and AhRKO mice. (**d**,**g**) Area under the curve (AUC) was measured to obtain glucose sensitivity for NCD (**d**) and HFD (**g**) groups. ** *p* < 0.001, *** *p* < 0.0001 by 2-way ANOVA with Tukey’s post hoc comparison.

**Figure 4 cells-12-01748-f004:**
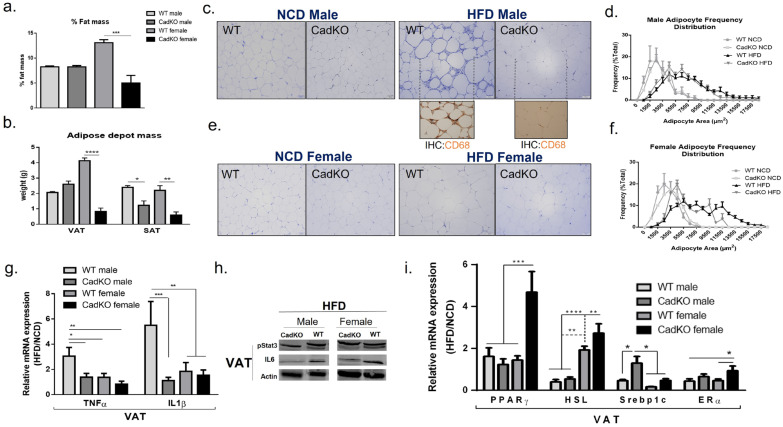
CadKO females experience a reduction in adipocyte size, while male mice experience inflammation. (**a**,**b**) % fat mass (**a**), visceral, and subcutaneous fat pad mass (VAT and SAT) (**b**) in CadKO (n = 3) and WT (n = 3) fed HFD for 15 weeks. (**c**,**d**) Histological data (H&E staining) obtained in VAT of CadKO and WT males (**c**) and females (**e**) after 15 weeks of NCD (n = 3) and HFD (n = 3–4). Scale bar: 50 μm. Immunohistochemistry (IHC) of CD68 antibody, marker for male adipose tissue macrophages on HFD males (**c**). Adipocyte area frequency distribution plotted for CadKO and WT group in males (**d**) and males (**f**). (**g**) Total RNA was isolated from VAT to investigate mRNA levels of TNFα and IL1β by real-time PCR on HFD male and female for CadKO (n = 4–6) and WT (n = 4–6) mice. (**h**) After 15 weeks of HFD feeding, VAT were harvested, and western blotting was performed to assess HFD-induced phospho-Stat3-IL6 pathway. β-actin was used to normalize. n = 3 for each group. Representative blots are shown for each group. (**i**) Total RNA was isolated from VAT to investigate mRNA levels of PPARγ, HSL, Srebp1c, and ERα by real-time PCR for CadKO (n = 4–6) and WT (n = 4–6) mice. Each gene was normalized against the amount of housekeeping gene, Actin. Data are expressed as the ratio of levels in HFD to NCD animals to accentuate changes associated with diet. * *p* < 0.05, ** *p* < 0.01, *** *p* < 0.001, **** *p* < 0.00001 by one-way ANOVA with Tukey’s post hoc multiple comparisons test.

**Figure 5 cells-12-01748-f005:**
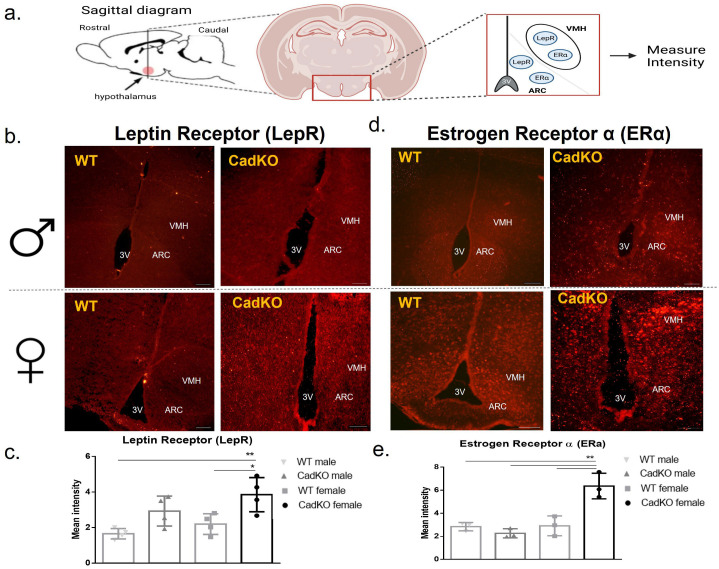
CadKO induces satiety response by promoting leptin signaling and estrogen signaling in female mice when fed according to a calorie-dense diet. (**a**) Diagrammatic illustration of the hypothalamus location in the sagittal section of the brain as well as location of third ventricle (3V), Arcuate (ARC) and ventromedial hypothalamic (VMH) nucleus in a coronal section to measure receptor intensity from immunofluorescence. (**b**,**d**) Immunofluorescence of (**b**) Leptin Receptor, LepR (red punctae) (**c**) Estrogen Receptor alpha, ERα (red punctae) expression in ARC and VMH nucleus located near 3V. Scale bar: 50 µm. (**c**,**e**) Mean intensity measured for LepR (**c**) and ERα (**e**) in CadKO (n = 4) and WT (n = 3) for both the sex, * *p* < 0.05, ** *p* < 0.01 by one-way ANOVA.

**Figure 6 cells-12-01748-f006:**
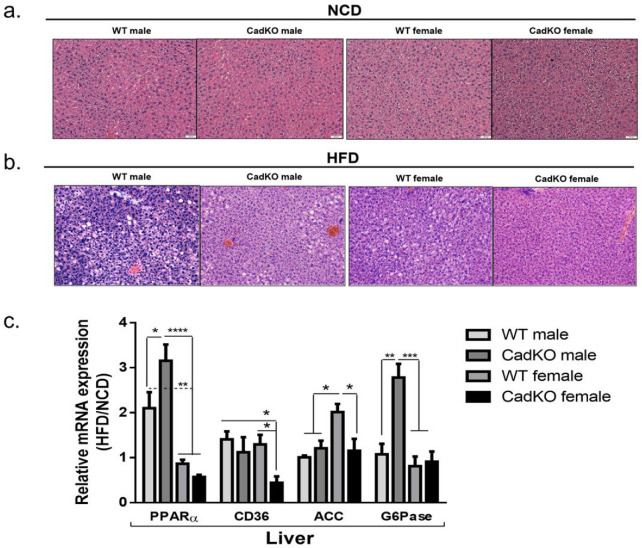
CadKO protection against HFD-induced hepatic damage is sexually dimorphic. (**a**,**b**) Histological data (H&E staining) obtained in the liver of the mice after 15 weeks of NCD (**a**) and HFD (**b**). White circular spots likely indicate the presence of lipid droplets. n = 3–4; Scale bar: 100 μm. (**c**) Total RNA was isolated from liver to investigate the mRNA levels of PPPARα, CD36, ACC, G6Pase by real-time PCR on male and female CadKO (n = 4–6) and WT (n = 4–6) mice. Each gene was normalized against the amount of housekeeping gene, Actin. Data are expressed as the ratio of levels in HFD to NCD animals to accentuate changes associated with diet. * *p* < 0.05, ** *p* < 0.01, *** *p* < 0.001, **** *p* < 0.0001 by one-way ANOVA with Tukey’s post hoc comparison.

**Figure 7 cells-12-01748-f007:**
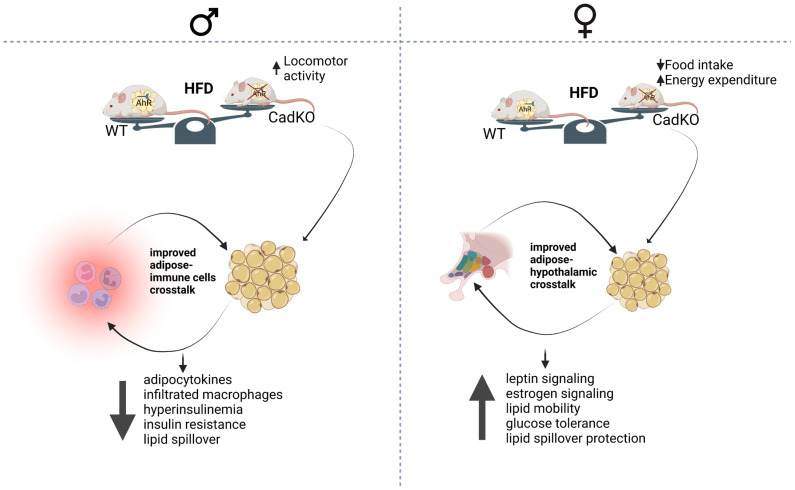
Illustration of sex-specific differences in CadKO mice for protection against HFD-induced metabolic dysfunction. CadKO females gained less weight due to net negative energy balance compared to WT, whereas CadKO males adapted via increased locomotor activity. Both the sexes for CadKO mice were observed to have improved adipose biology, which helped females to maintain a healthy adipose–hypothalamic network, whereas CadKO males maintained healthy adipose–immune cell crosstalk. Created by Biorender.com.

**Table 1 cells-12-01748-t001:** Serum measurements.

	NCD				HFD			
	CadKO Male	WT Male	CadKO Female	WT Female	CadKO Male	WT Male	CadKO Female	WT Female
Insulin (ng/mL)	0.34 ± 0.22 ^#^	0.81 ± 0.30 ^#^	0.46 ± 0.34	0.63 ± 0.34	1.59 ± 0.46 ^$^	6.46 ± 5.14 ^$^	0.49 ± 0.34	0.88 ± 0.54
Leptin (ng/mL)	7.76 ± 2.64 ^#^	5.27 ± 3.88 ^#^	4.77 ± 2.57 ^#^	5.56 ± 1.01 ^#^	40.40 ± 2.76 ^$^	43.47 ± 2.40	28.81 ± 11.19	38.67 ± 3.19
Adiponectin (mg/mL)	48.52 ± 10.38	35.85 ± 6.25 ^$,#^	75.75 ± 18.9	71.52 ± 31 ^#^	71.42 ± 9.64	73.90 ± 7.96 ^#^	78.15 ± 24.22	106.85 ± 26.6
17β-estradiol (pg/mL)	155.10 ± 1.99	155.73 ± 12.4 ^$,#^	161.39 ± 13.4	178.8 ± 16.63	144.76 ± 24.92	125.74 ± 13 ^$^	161.48 ± 11.5	160.13 ± 6.57
Triglycerides (mg/dL)	n/a	n/a	n/a	n/a	195.88 ± 26 ^$^	181.74 ± 6.23	155.33 ± 17.78	173.20 ± 5.68

Abbreviations: CadKO: Conditional adipose Knock Out; WT: Wild-Type; HFD: high-fat diet; NCD: normal chow diet. Blood was collected from mice groups at week 15 of NCD or HFD. Serum leptin, insulin, adiponectin, 17β-estradiol and triglycerides were measured by ELISA or commercial kits (NCD, n = 5–6; HFD, n = 8–10). ^#^ *p* < 0.05–0.0001, compared across diets within sex; ^$^ *p* < 0.05–0.0001, compared across sex within diet, by two-way ANOVA with Turkey’s post hoc comparison. n/a, no measurements were taken.

## Data Availability

The data presented in this study are available on request from the corresponding author.

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
