# Peer review of "Deficiency of Adipose Aryl Hydrocarbon Receptor Protects against Diet-Induced Metabolic Dysfunction through Sexually Dimorphic Mechanisms"

_cells, 2023, doi:10.3390/cells12131748_

Round 1
Reviewer 1 Report
Summary
The submitted manuscript aims to test the hypothesis that conditional knockout of the AHR from mature adipose tissue will improve metabolic dysfunction from treatment with a high fat diet. To accomplish this an Adionectin-CreERT2::Ahrfx/fx::Tomato/+ mouse model model was fed either control diet of high fat diet for 15 weeks and monitored for various metabolic parameters including calorimetry, activity, glucose handling, and immunohistochemical analyses. The authors show that CadKO mice showed either reduced (males) or no (females) weight gain on HFD and that glucose tolerance was improved in females. Moreover, in female mice CadKO were found to have increased sensitivity to leptin and activity of estrogen receptor signaling while male mice showed evidence of delayed obesity and insulin resistance. Collectively, these studies show sexually dimorphic function of adipocyte AHR resulting in an overall protection against metabolic dysfunction.
Comments:
· Can the authors verify that Figure 6 is accurately described. The resolution is poor but it seems like panel A and B use different staining protocols or maybe tissues were fixed differently? Red blobs (or what might appear as blobs at this resolution) almost look like Oil Red O droplets but the remaining tissue would suggest this is not the case. The difference is quite noticeable so it may be important to note why there is such a difference.
· Lines 405 – 407: Normalization of triglyceride levels to body weight seems abnormal. Can the authors point to citations that would justify this measurement as a putative measure of increased lipid mobility? This endpoint would have a much greater impact with evidence for its representation of a biologically meaningful endpoint.
· It is recommended that the authors examine their figures for ways to improve the readability which would be greatly beneficial to readers. For example:
o Use a consistent symbol or color for each model (CadKO, AhrKI, WT) and then a consistent symbol or color for NCD or HFD in panels B/C (Figure 2).
o Ensure that the reference group (control or other appropriate reference) is always first in barplots and histology figures. This seems to be the case until Figure 4 where CadKO becomes the first data shown.
o Sometimes males are shown first, other times females are shown first. Consistency here would also significantly improve the readability.
· Table 1: The column indicates “Serum Protein” but also includes a metabolite, triglyceride. The table footnote should state what n/a means (not measured, not detected, …?). On the same note, methods indicates the triglyceride assay under ELISA which is not technically correct.
· Line 190: … with primary antibodies (listed below). Should it not be pointing to supplemental data?
· Materials and methods doesn’t seem to include methods relevant for supplemental data. How was consumption, calorie intake, and body temperature measured?
· Numerous combinations of “*”, “#”, and “$” are used to denote significance from p < 0.05 to P < 0.00001. The authors should consider whether the value added from distinguishing P < 0.05 vs. P < 0.001 and others is really critical. Figures and tables could be greatly simplified and easier to interpret without as many symbols.
· Line 411: Srebp1c is not generally considered a rate-limiting enzyme as it is a transcription factor that targets rate-limiting enzymes for lipid metabolism.
Reviewer 2 Report
Haque and colleagues studied the effects of an inducible adipocyte specific knockout of AhR (CadKO) in the context of lean and DIO and assessed both sexes, and include a global congenital AhR KO as a comparison. The results suggest that ablation of adipocyte AhR is protective against DIO and several associated metabolic disease outcomes. The rationale for study is strong and their findings are novel and interesting. However, I have several concerns that must be addressed (see below)
1. The majority of images in the main document are of low or poor quality. For example in Figure 1, the immunofluorescent images in panel a. are out of focus. and panels b. and d. are fuzzy and appear to have been stretched. This is not the only figure that needs improvement, many are not publication quality. Additionally many of the images are too small to make out the finer details.
2. At times the denotation of significance is unclear as to what is being compared. For example figure 3g, are these statistical significant notations in comparison to the WT group, or are some comparing the two KO groups? In relation to this there are some statistical findings that are notated and discussed that are not meaningful comparisons, such as in Table 2 where the HFD male WT group is compared to the female CadKO group.
3. The legend and graph order need to be consistent throughout. Along the same lines, sometimes the males and females are separated with individual graphs and sometimes together. Please provide consistent reporting of findings for ease and reduced confusion for thee reader. Males and females should b plotted separately in my opinion.
4. There are several areas in the text where the font or text size is different.
5. Please provide additional details in the methods regarding the parameters for the immunofluorescent images-- were all settings such as exposure time etc., kept consistent across images for the groups.
6. H and E staining can give an idea about lipid droplets/steatosis, but the appearance of white areas does not always reflect lipid accumulation and the authors cannot conclude steatosis is present or attenuated based on this alone. Lipid quantification or actual lipid staining such as Oil Red O or nile red would need to be performed. Please include one of those measures or revise your conclusions to reflect this.
7. In figure 4, why are the histological images grey? Please include the H & E images and the CD68 staining for all.
8. Although the HFD-fed WT males weigh much more in total than the females, the females have greater fat pad weights. Please comment/clarify.
9. There is entirely too much unnecessary information in the introduction, it reads more like a review, please revise for concision and include only what is relevant.
10. The discussion is disorganized and quite long with some redundancies please also edit this section for concision.
11. Please rephrase or rewrite the following for clarity:
Lines 50-51
Line 297
12. Would the authors please justify such a prolonged fasting period for the GTT.
13. Consider reporting the p value in the text in the results when reporting significant findings.
Overall the English is good, with a few grammatical errors here and there which are discussed in more detail in my comments to the authors.
Round 2
Reviewer 2 Report
I appreciate the authors addressing my comments from the previous review. The figures are much improved and it reads better overall. However, several concerns remain. Please see below for more information.
1. It appears that table 1 was removed and not replaced
2. Why are the H and E stains blue, it is usually a pink color? When they were grey scale you could see the outline of the individual adipocytes better, but it should be the color of the stain and still be visible.
3. Figure 5b,d-- the image and immunofluorescence staining appear to be identical when comparing LepR and ERa images in the female CadKO. It seems that it is the exact same image with all of the bright spots identical, but is slightly stretched for ERa, you can even see the enlarged scale bar. The graphs are different. This was harder to notice with the first submission with the poor image quality, but I went back to look and they do appear the same as well. This is a major issue that needs to be addressed.
4. I still find the findings overstated in the results/discussion in regards to liver steatosis, while I agree the histology may be suggestive of liver fat, that is not always the case and hepatic lipid needs to be measured or the results should be rephrased to address that it was not directly measured.
Minor:
1. On page 11 lines 35-36, there is text missing at the beginning of the sentence
2. Figure 3, panel A-- consider separating males and females and place beside the other individual graphs below so that this graph does not need to be enlarged and everything fits nicely.
3. The yellow text on the histology images in figure 4 does not print well--consider changing to a more distinctive color.
4. Figure 4 is covering some of the text below it.
5. Figure 6 legend is in the wrong place
Author Response
I appreciate the authors addressing my comments from the previous review. The figures are much improved and it reads better overall. However, several concerns remain. Please see below for more information.
- It appears that table 1 was removed and not replaced
Table 1 shows up in my version of the previous submission. At any rate, it is in the updated version as well.
- Why are the H and E stains blue, it is usually a pink color? When they were grey scale you could see the outline of the individual adipocytes better, but it should be the color of the stain and still be visible.
I think you can find various shades of pink, purple and blue throughout the literature. We had some difficulty with background in the adipose H&E staining, which is why we preferred to go with gray scale images, especially since the only purpose of these images is to show the adipocytes size. In response to the previous critiques, we switched the images out to color. Regarding the pink color intensity in the H&E staining, we understand that the typical expectation is for the cytoplasm to exhibit a pink hue due to the eosin stain. We encountered some challenges due to the nature of the tissue itself. Adipose tissue is inherently squishy, which posed difficulties in obtaining sections of consistent thickness. In order to maintain standardization across all adipose tissue samples, we opted for a section thickness of 10 microns. This requires a longer eosin exposure time than usual, but we decided to adhere to our standard protocol in order to maintain consistency. Our aim was to ensure that the staining procedures were as comparable as possible between different tissue types, facilitating accurate comparisons and analysis. Given the focus of our study on the overall tissue morphology and the specific investigation of crown-like structures, this staining protocol effectively served our research objectives. Moreover, there were some articles we found that looks similar to our images, and are cited below:
Roerink SHPP, Wagenmakers MAEM, Langenhuijsen JF, et al. Increased Adipocyte Size, Macrophage Infiltration, and Adverse Local Adipokine Profile in Perirenal Fat in Cushing's Syndrome. Obesity (Silver Spring). 2017;25(8):1369-1374. doi:10.1002/oby.21887 (Figure 1)
Mccourt, A.C., Jakobsson, L., Larsson, S., Holm, C., Piel, S., Elmér, E., Björkqvist, M., 2016. White Adipose Tissue Browning in the R6/2 Mouse Model of Huntington’s Disease. PLOS ONE 11, e0159870.. https://doi.org/10.1371/journal.pone.0159870 (Figure 1)
- Figure 5b,d-- the image and immunofluorescence staining appear to be identical when comparing LepR and ERa images in the female CadKO. It seems that it is the exact same image with all of the bright spots identical, but is slightly stretched for ERa, you can even see the enlarged scale bar. The graphs are different. This was harder to notice with the first submission with the poor image quality, but I went back to look and they do appear the same as well. This is a major issue that needs to be addressed.
Thank you for bringing this error to our attention. The necessary corrections have been made. We appreciate your valuable feedback and have revised the figure accordingly.
- I still find the findings overstated in the results/discussion in regards to liver steatosis, while I agree the histology may be suggestive of liver fat, that is not always the case and hepatic lipid needs to be measured or the results should be rephrased to address that it was not directly measured.
We have backed off further in our description of these findings. Hepatic fat was not directly measured in this study, although we have previously measured it in our studies on the global AhRKO mice. We don’t think that adding oil red staining will add much to our study, so prefer to leave it as is.
Minor:
- On page 11 lines 35-36, there is text missing at the beginning of the sentence
I read all of page 11 and cannot find this problem.
- Figure 3, panel A-- consider separating males and females and place beside the other individual graphs below so that this graph does not need to be enlarged and everything fits nicely.
Thank you for bringing this concern to our attention. The necessary corrections have been made. We appreciate your valuable feedback and have revised the figure accordingly.
- The yellow text on the histology images in figure 4 does not print well--consider changing to a more distinctive color.
This has been changed according to the reviewer’s suggestion.
- Figure 4 is covering some of the text below it.
Figure 4 appears ok to me.
- Figure 6 legend is in the wrong place
This is corrected.

Round 3
Reviewer 2 Report
I thank the authors for addressing my comments. I find the manuscript to be much improved. The only remaining issue is that for the histology images in figure 5b and 5d, the scale bars appear to not be identical in size when comparing the groups.
Author Response
One of the images was cropped slightly differently than the others. It has been fixed. The scale bar is imbedded into the image when the image is taken with the microscope camera. So data were not altered, only the size of the picture.